# Human Fall Detection with Ultra-Wideband Radar and Adaptive Weighted Fusion

**DOI:** 10.3390/s24165294

**Published:** 2024-08-15

**Authors:** Ling Huang, Anfu Zhu, Mengjie Qian, Huifeng An

**Affiliations:** School of Electrical Engineering and Information Engineering, Lanzhou University of Technology, Lanzhou 730050, China; 222085406038@lut.edu.cn (A.Z.); 232085407025@lut.edu.cn (M.Q.); 232081102006@lut.edu.cn (H.A.)

**Keywords:** fall behavior recognition, adaptive weighted fusion, UWB radar

## Abstract

To address the challenges in recognizing various types of falls, which often exhibit high similarity and are difficult to distinguish, this paper proposes a human fall classification system based on the SE-Residual Concatenate Network (SE-RCNet) with adaptive weighted fusion. First, we designed the innovative SE-RCNet network, incorporating SE modules after dense and residual connections to automatically recalibrate feature channel weights and suppress irrelevant features. Subsequently, this network was used to train and classify three types of radar images: time–distance images, time–distance images, and distance–distance images. By adaptively fusing the classification results of these three types of radar images, we achieved higher action recognition accuracy. Experimental results indicate that SE-RCNet achieved F1-scores of 94.0%, 94.3%, and 95.4% for the three radar image types on our self-built dataset. After applying the adaptive weighted fusion method, the F1-score further improved to 98.1%.

## 1. Introduction

According to a 2018 survey by the World Health Organization (WHO), there are approximately 60,000 fatal fall incidents worldwide each year, the majority of which involve elderly individuals aged 65 and older [1]. Severe falls can lead to serious consequences such as fractures, dislocations, head injuries, and sprains [2]. In the member states of the European Union, injury issues affect approximately 105,000 people. Additionally, nearly 40,000 elderly individuals are declared dead due to falls. In the United States, the fall-related mortality rate among people aged 60 and older is 36.8 per 100,000, whereas in Canada, it is 9.4 per 10,000 for the same age group. In Australia, Canada, and the United Kingdom of Great Britain and Northern Ireland, the hospitalization rate for injuries due to falls among those aged 60 and older ranges from 1.6 to 3.0 per 10,000 citizens. In Western Australia and the United Kingdom, this rate is as high as 5.5 to 8.9 per 10,000 people [3]. Accurate identification of fall types can assist medical professionals in quicker diagnosis and formulation of effective treatment plans. Furthermore, integrating this technology with smart homes and health monitoring systems can significantly enhance the quality of life for the elderly and other vulnerable groups. The application of such recognition technology can facilitate more targeted treatment measures, reduce the risk of secondary injuries caused by falls, and promote the development of public health strategies and personalized medicine [4,5].

Currently, Human Motion Recognition (HMR) primarily relies on cameras. However, this method tends to invade personal privacy and requires unobstructed and well-lit conditions [6]. Although wearable devices overcome visual limitations by using inertial sensors or plantar pressure sensors, these devices may impose physical burdens, especially for the elderly and visually impaired individuals [7]. In contrast, environmental recognition methods using WiFi, infrared sensors, or radar can detect activities without adding burdens to the subjects or invading their privacy, making them the most ideal recognition methods currently available [8,9,10,11].

In previous studies, Zhao et al. addressed the issue of optimizing video images through network backpropagation by replacing time-frequency diagrams with convolutional layers and max-pooling layers. They also utilized fusion layers and an attention mechanism-based encoder-decoder structure to fully exploit the temporal information in the raw data, achieving effective recognition of continuous behaviors [12]. Li et al. proposed an enhanced contrastive learning dual transformer method (SCL-SwinT). This method incorporates overlapping patch embeddings and cosine similarity attention in the dual transformer encoder, combined with supervised contrastive learning (SupCon). By integrating two patch-embedding methods (ResNet and MLP) and three patch-partitioning strategies (sequential, grid, and strip), the model can extract contextual information from different angles and depths, achieving over 90% recognition accuracy on the IURHA2023 dataset [13]. Chen proposed a human activity classification method based on multiconvolutional neural network (CNN) information fusion. Six different data preprocessing methods were selected, and the results were input into corresponding CNN models for parameter training. The weighted voting method was used for information fusion to obtain the final classification results [14].

Fall behavior recognition is a specialized branch of human behavior recognition. Compared to general human behavior recognition, fall behavior recognition poses greater challenges in terms of recognition difficulty and data collection. Hemi et al. designed a radar map deep learning fusion network based on MobileNet-V3, significantly improving fall recognition performance. However, their consideration of spectral weight distribution was relatively limited, and they did not further classify fall types [15]. He et al. used parallel 2DCNNs to extract features, optimized by CBAM, to achieve feature-level fusion, but their study was limited to distinguishing single fall actions from various non-fall actions, with a limited range of action types in the experimental dataset [16]. Yao et al. proposed a fall detection system based on millimeter-wave radar, achieving high accuracy and robustness through neural network and information fusion technologies. This system collected target distance, speed, and angle information using FMCW radar, generating range-speed maps, range-horizontal angle maps, and range-vertical angle maps through three independent neural networks. To improve detection accuracy, this study used an ensemble learning stacking method to fuse the features extracted by the three neural networks. Although this study trained and recognized using a large dataset containing various action types, it mainly focused on accurately detecting fall events and distinguishing non-fall actions, rather than further classifying different types of falls [17].

Building on the work of Hemi et al., this paper refines the classification of fall behavior recognition and improves decision-level fusion methods. By embedding SE modules into Dense Residual Blocks composed of dense and residual connections, we achieved adaptive adjustment of feature importance, enhancing the recognition of fall behaviors and their types. Through in-depth comparisons with other advanced neural network models, the superiority of this network has been demonstrated. Additionally, this paper proposes an adaptive weighted fusion method to weight training results, improving the accuracy of behavior recognition.

The innovations of this paper are summarized as follows:Novel Deep Learning Feature Extraction Network SE-RCNet: This network combines spatial attention mechanisms and residual connections, significantly enhancing the ability to extract key features from radar images, improving target detection and behavior classification performance under complex environmental conditions.By employing a differentiated weight distribution mechanism, different weights are assigned to each radar map feature based on their contribution and impact in the decision-making process. By comprehensively evaluating the influence of each radar map feature, this method can more accurately determine behavior types.

The remainder of this paper is organized as follows: Section 2 describes the design process of the self-built dataset and introduces the public datasets used. Section 3 provides a brief overview of the adaptive weighted fusion network method and its structural diagram. Section 4 presents the experiments and results analysis. Section 5 briefly summarizes the content of this paper and discusses prospects for future research.

## 2. Dataset

### 2.1. Self-Built Dataset

In our experiment, we used the PulsOn 440 radar module from Time Domain. This UWB radar has a center frequency of 4.3 GHz and a frequency range of 3.1 to 4.8 GHz. The sampling frequency is 16.387 GHz. The antenna’s RF radiation intensity complies with FCC Part 15 regulations in the United States and adheres to ETSI EN 302065 standards in Europe [18]. We set the pulse repetition frequency (PRF) to 240 Hz to achieve higher velocity resolution, enabling the radar to detect finer movements. This is crucial for accurately identifying human positions and movements. The main radar parameter settings are shown in Table 1.

#### 2.1.1. Dataset Description

In this study, we collected a dataset containing human fall behaviors using the PulsOn 440 UWB radar module (As shown in Figure 1) in an indoor environment. During the experiments, the radar was installed at a height of 1 m, and all experiments were conducted within the effective range of the radar to ensure data consistency and validity. This dataset includes fall behavior echo data from 9 subjects (2 females and 7 males). The basic information of the participants is shown in Table 2. Due to the inclusion of participants of different genders (male and female) and varying body profiles (height and weight), although they perform the same actions, the range of their movements is not entirely identical. This diversity allows us to simulate a wide range of human movements, ensuring that the collected data cover a broad spectrum of real-world scenarios.

Each subject performed data collection for 10 different types of behaviors. These behaviors include five types of daily activities: sitting down, bending to pick up an object, standing up, walking towards the radar, and walking away from the radar; as well as five types of fall behaviors: falling while sitting down or standing up, falling sideways to the radar, falling backwards to the radar, falling at a 45-degree angle to the right of the radar, and falling at a 45-degree angle to the left of the radar. Each subject provided 30 samples for each behavior type, and a total of 2700 samples were collected for classification and recognition research. Detailed action types and descriptions are shown in Table 3:

#### 2.1.2. Data Preprocessing

As shown in Figure 2, to filter out stationary or slowly moving interference targets, the raw data collected by the radar are first processed using Moving Target Indication (MTI). We use a three-pulse canceller to eliminate the echoes of stationary targets by comparing the phase differences of consecutive pulses. The differential processing formula is as follows:(1)y(n)=x(n+1)−2x(n)+x(n−1)

This formula effectively suppresses the echo signals of stationary targets by performing a weighted summation of the current pulse and the two adjacent pulses, thereby highlighting the signals of moving targets. This allows subsequent processing and analysis to focus more on moving targets. However, differential processing may lead to partial loss of information on stationary or slowly moving targets, as it attenuates slowly varying signal components, causing this information to be partially or completely lost. To address this, we introduced the Hilbert transform in subsequent processing. By generating an analytic signal, it provides instantaneous attributes (such as instantaneous phase and frequency) to preserve the complete information of the target as much as possible. The Hilbert transform helps to more comprehensively retain the characteristics of the signal, especially in frequency analysis and dynamic signal processing. It reduces artifacts and improves the accuracy of signal processing, thereby mitigating the distortion and information loss caused by differential processing [19]. The generated range-time diagram is shown in Figure 3a.

Subsequently, the Short-Time Fourier Transform (STFT) is applied to obtain the spectral information for each time window. In STFT, the signal is divided into multiple time windows. The signal within each window undergoes a Fourier transform, and through integration (i.e., summation), the intensity of each frequency component is calculated. This allows us to obtain the distribution of the signal across time and frequency. The generated time–doppler diagram (as shown in Figure 3b) shows the Doppler shift of the target at different time points, reflecting the speed changes in the target. The choice of time window size and overlap parameters in STFT directly affects the spectral resolution and time resolution. Inappropriate parameters may lead to spectral information distortion. To avoid this, we experimentally determined that the optimal window size is 128 points and the overlap parameter is 110 points. A window length of 128 points can capture important frequency components in radar signal preprocessing while maintaining reasonable time resolution. An overlap of 110 points improves the time resolution, making the spectrogram smoother and more continuous, reducing information loss caused by edge effects. These parameter choices are based on multiple experimental results, providing an optimal balance between frequency and time resolution in radar signal preprocessing. This ensures that the generated images have smooth time resolution and spectral continuity, thereby reducing signal distortion.

Finally, a Fast Fourier Transform (FFT) is performed on the time-range map data to generate a range–doppler diagram (as shown in Figure 3c). The range–doppler diagram can display the distribution of targets at different distances and frequency. By analyzing the color differences in these three types of radar images, we can clearly identify the interrelationships among distance, time, and Doppler frequency features, thereby providing more comprehensive target information and a reliable data source for subsequent decision fusion.

### 2.2. Public Dataset

To further demonstrate the generalization ability of the model, this study also utilized the K-band ultra-wideband radar fall detection dataset published by the Journal of Radars. This dataset includes fall and non-fall behavior data collected in various test environments, covering participants of different ages, heights, and genders, thus exhibiting significant diversity and representativeness. By incorporating this dataset, the model’s performance under different conditions and among diverse populations can be comprehensively evaluated.

This study primarily used ten different types of fall behaviors from this dataset to evaluate the network model’s effectiveness in recognizing similar fall behaviors. By using these ten types of falls, we can verify whether the model can maintain high recognition accuracy when dealing with highly similar fall actions. These fall behaviors are meticulously recorded in different environments and scenarios, providing rich samples for model training and validation. The detailed action types are shown in Table 4, covering various typical fall scenarios to comprehensively assess the model’s fall detection capabilities. The inclusion of this public dataset not only enhances the credibility and generalizability of the research results but also provides a solid data foundation and reference for subsequent research.

## 3. The Adaptive Weighted Fusion Network Recognition Method

### 3.1. Model Construction

This paper proposes a human fall behavior recognition model based on SE-RCNet. The model takes 80 × 80 pixel radar images as input and significantly improves the accuracy of fall behavior classification through a series of key mechanisms.

First, the model captures the basic features of radar images through an initial extraction module composed of convolutional layers and ReLU activation functions. Then, six densely connected residual blocks (labeled as block1 to block6 in the diagram) further refine and process these features. Each residual block consists of two convolutional layers, batch normalization, and ReLU activation functions, which not only enhance the ability to capture nonlinear features but also improve information flow and feature retention through the residual learning mechanism.

Each residual block integrates a Squeeze-and-Excite (SE) module. This module first compresses the spatial dimensions through global average pooling and then performs channel recalibration, adaptively adjusting the weights of the feature channels, thereby making the model focus more on important features and improving classification performance [20].

The densely connected residual blocks achieve residual learning through element-wise addition, where the output of the second convolutional layer is added to the output of the first convolutional layer, promoting information flow and feature retention in the deep network. The outputs of each residual block are merged to ensure the complete transmission of information between the deep layers while enriching feature representation.

After each densely connected residual block, a max-pooling layer further reduces feature dimensions and computational load, providing higher-level abstract feature representations for the model. Finally, a flattening layer converts the multidimensional feature maps into one-dimensional feature vectors, which are then input into the fully connected layer to achieve fall behavior classification.

The overall network structure is shown in Figure 4.

### 3.2. Adaptive Weighted Fusion

#### 3.2.1. Confidence

In classification tasks, confidence measures the certainty of the model’s predictions. The model assigns a probability value to each category, with the sum of the probabilities for all categories equaling 1. The highest probability value represents the category that the model considers most likely. In short, the category with the highest probability is the model’s most confident prediction.

#### 3.2.2. Decision-Level Fusion Network

The original decision fusion network was designed for binary classification problems of fall versus non-fall and is not suitable for multiclass classification problems. Therefore, we have modified it. The results of the three types of radar images are calculated and aggregated. If two or more radar images produce the same output, that output is used directly. If all outputs are different, the confidence of each output is compared, and the result with the highest confidence is chosen, as shown in Figure 5.

Although we have optimized the original decision fusion network to some extent, it still has a critical flaw: the network implicitly assigns equal weights to the results of the three different radar images. This approach is clearly inconsistent with our radar image recognition results. In reality, the recognition accuracy of these three radar images is different. Using traditional decision fusion methods not only diminishes the impact of the radar image with higher recognition accuracy but also amplifies the shortcomings of the radar image with lower recognition accuracy.

#### 3.2.3. The Adaptive Weighted Fusion Network

In this paper, we adopted an adaptive weighted fusion method to enhance the performance of the decision-level network. This is an efficient strategy for addressing behavior recognition problems in multiclass tasks based on confidence levels. In this process, the confidence of each class prediction, which is the probability value output by the model, plays a central role. For the recognition tasks of each type of radar image, we no longer use the highest confidence principle alone. Instead, we employ an adaptive weighting mechanism that dynamically adjusts their influence in the final fusion decision by considering the recognition performance of different radar images on the validation set.

Specifically, the weights in the fusion network are determined by the accuracy of each individual radar image. When dealing with radar images of different accuracies, our fusion model automatically increases the weight of the more accurate results and reduces the weight of the less accurate results. This means that the model will synthesize and select the most reliable result as the final output based on the confidence and adjusted weight of each radar image.

The advantage of this adaptive weighting strategy is that it can automatically balance and optimize the integration of multisource information without sacrificing recognition accuracy. Our model applies this strategy to the SE-RCNet architecture, achieving more precise behavior recognition in the radar image dataset. Particularly in distinguishing different types of fall behaviors, this method not only improves recognition accuracy but also enhances the model’s adaptability and robustness under varying conditions.

#### 3.2.4. The Adaptive Weighted Fusion Network-Specific Steps

(1) Dataset Partitioning and Initial Model Training

To ensure the fairness and scientific validity of model training and evaluation, this study first randomly divides the collected dataset into training, validation, and test sets in a 6:2:2 ratio. This step not only ensures comprehensive and random data coverage but also provides a rigorous benchmark for model evaluation.

After training the SE-RCNet model on the training set, we evaluate the initial performance of the model using the validation set and obtain the initial accuracy for various action categories in the fall type recognition task. This accuracy reflects the model’s ability to recognize unseen data.

(2) Radar Map Type Weight Allocation

The model trained with the training set was used to evaluate the validation set, yielding the initial radar map recognition accuracy n = [n_1_, n_2_, n_3_], where n_k_ represents the accuracy of the kth radar map type. Then, each sample in the test set was input into the model. For each individual sample, the model outputs logits for ten categories. These logits are then input into the softmax function, which outputs the predicted probability for each category. The highest probability indicates the model’s confidence.

Assume the test set outputs a logits vector z = [z_1_, z_2_, …, z_10_], where z_i_ is the magnitudes of logits for various categories in the sample. The softmax function converts this vector into another vector s = [s_1_, s_2_, …, s_10_], where s_i_ is calculated as follows:(2)$$s_{i}=\frac{e^{z_{i}}}{\sum_{j=1}^{n}e^{z_{j}}}$$

In this expression, $e^{z_{i}}$ is the natural exponent of $z_{i}$. The denominator $\sum_{j=1}^{n}e^{z_{j}}$ is the sum of the natural exponents of all $z_{j}$ elements. n is the total number of classes, which is 10. The result of the softmax function is a probability distribution over the categories, where each element’s probability value is between 0 and 1, and the total sum of probabilities is 1.

To find the model’s confidence and its category, we use the following formula: the maximum probability $s_{\max}=\max\left(s\right)$, where $s_{\max}$ = max(s) indicates the highest value in the vector s. The maximum category index $i_{\max}=\mathrm{argmax}\left(s\right)$, where argmax(s) is the index of the maximum value $s_{\max}$ in the vector s. The three radar maps’ $s_{\max}$ values for each test set sample are normalized to obtain the normalized probability matrix $m=[m_{1},\;m_{2},{\;m}_{3}]$. 

(3) Weighted Test Evaluation

We perform the Hadamard product of the probability output n from the validation set and the highest class recognition rate m from each test set spectrum to obtain the final probability-adjusted vector w = [$w_{1}$,$w_{2}$,$w_{3}$], where $w_{i}$ represents the adjusted highest class recognition rate of the i-th spectrum. The specific calculation formula is expressed as follows:(3)w=n∘m

Then, we sum the probabilities of the same class in W to obtain a new probability vector w’. We use $i_{\max}=\mathrm{argmax}(w’)$ as the final output class of the fusion network.

The overall recognition flowchart is shown in Figure 6.

## 4. Experimental Results and Analysis

### 4.1. Experimental Environment Setup

The computational environment for this study is configured as follows: The computer is equipped with an Intel i7-12700 processor, 32 GB of memory, and an NVIDIA GeForce RTX 3070 graphics card. The development environment is based on Anaconda 3, using Python 3.9.18. The construction, training, and performance evaluation of the deep learning models are carried out using TensorFlow 2.10.0 and Keras 2.7.0.

### 4.2. Comparative Experiment

In this section, we compare the proposed SE-RCNet network and adaptive weighted network with current mainstream deep learning networks and fusion methods on both the self-built dataset and the public dataset. Due to the specificity of the fall behavior recognition task, missing important positive samples (i.e., failing to detect fall behaviors) has a more severe impact than misclassifying samples (i.e., incorrectly detecting non-fall behaviors as fall behaviors). Therefore, in addition to using accuracy as a metric, more evaluation metrics are needed to comprehensively measure the overall performance of the network.

To this end, we introduce precision, recall, and F1-score as evaluation metrics in addition to accuracy. Among these metrics, TP (true positive) represents the number of times the model correctly identifies fall behaviors, TN (true negative) represents the number of times the model correctly identifies non-fall behaviors (i.e., the total number of correctly identified non-fall behaviors), FN (false negative) represents the number of times the model fails to identify fall behaviors (i.e., incorrectly identifying fall behaviors as other behaviors), and FP (false positive) represents the number of times the model incorrectly identifies fall behaviors (i.e., incorrectly identifying other behaviors as fall behaviors). The F1-score is a comprehensive metric that considers both precision and recall, providing a more complete assessment of the model’s performance.
(4)$$\mathrm{Accurac}=\frac{\mathrm{TR}+\mathrm{TN}}{\mathrm{Total}\;\mathrm{Cases}}$$
(5)$$\mathrm{Precision}=\frac{\mathrm{TP}}{\mathrm{TP}+\mathrm{FP}}$$
(6)$$\mathrm{Recall}=\frac{\mathrm{TP}}{\mathrm{TP}+\mathrm{FN}}$$
(7)$$\text{F1-score}=2\times \frac{\mathrm{Precision}\times \mathrm{Recall}}{\mathrm{Precision}+\mathrm{Recall}}$$

The system studied in this paper consists of two main subsystems: the radar image preliminary recognition network and the fusion optimization method. Since these two systems are technically progressive, we will evaluate the performance of each system separately.

First, we will assess the performance of the radar image preliminary recognition network. This subsystem is primarily responsible for extracting features from radar images and performing initial recognition. Then, we will evaluate the performance of the fusion optimization method. This subsystem optimizes and fine-tunes the initial recognition results, thereby improving overall recognition accuracy and robustness.

On this basis, we will compare the system studied in this paper with other existing networks to comprehensively evaluate its performance in fall behavior recognition. Through this phased evaluation and comparative analysis, we can gain a deep understanding of the advantages and disadvantages of each subsystem and verify the effectiveness of our system in handling highly similar fall behaviors.

#### 4.2.1. Network Model Comparison

This paper compares the recognition performance of different deep learning models in fall detection. The selected comparison networks include CNN-4, DenseNet121 [21], ResNet50 [22], and Xception [23].

CNN-4 is a basic convolutional neural network structure with fewer layers and parameters, making it suitable as a baseline comparison model. DenseNet121, due to its densely connected layers, promotes gradient flow and feature reuse, thereby improving the overall performance of the model. ResNet50 addresses the degradation problem in deep neural networks, demonstrating excellent performance and stability, and is considered a classic network in the field of deep learning. Xception, by adopting depthwise separable convolutions, significantly reduces computational complexity while maintaining efficient model performance. These models represent different deep learning architectures and design philosophies. By comparing their performance, we can comprehensively evaluate the superiority of the SE-RCNet model in radar image processing and behavior recognition tasks. Table 5 lists the recognition rates of all networks.

Figure 7 shows the cross-entropy loss values (as in Figure 7a,c,e) and accuracy (as in Figure 7b,d,f) curves with increasing training epochs using the SE-RCNet network for five-fold cross-validation on individual spectrograms. As seen from the figures, after 100 training iterations, the accuracy and cross-entropy loss values for all three spectrograms tend to stabilize. Table 5 lists the average results of five-fold cross-validation on the three spectrograms for each network. Experimental data indicate that our proposed SE-RCNet model performs excellently in radar spectrum feature extraction and action recognition tasks, outperforming other comparison models across multiple evaluation metrics. 

As a classic image recognition network, ResNet50 also performs quite well on our custom dataset, second only to SE-RCNet. However, the performances of CNN-4, DenseNet121, and Xception networks on various metrics are slightly inferior to those of SE-RCNet and ResNet50.

To evaluate the performance differences of different deep learning models in the fall detection task, this study employed one-way analysis of variance (ANOVA), which is used to test whether there are significant differences among multiple sample means. In this study, we considered the models as the factor and the F1-scores of the models as the response variable. Statistical analysis was conducted using the SciPy library. This statistical method helps determine whether there are significant differences in accuracy among the models, providing scientific basis for model selection. The results show significant differences in F1-scores among the models across the three types of radar spectrograms. Specifically, for the time–distance spectrogram (F(4, 20) = 36.618, *p* < 0.001), the time–distance spectrogram (F(4, 20) = 115.425, *p* < 0.001), and the range–distance spectrogram (F(4, 20) = 38.679, *p* < 0.001), all results indicate that the differences are statistically significant. Additionally, we conducted post hoc tests using Tukey’s HSD test to examine the differences between each pair of models. The post hoc test results are shown in Figure 8.

From the recognition results on the public dataset shown in Table 6, it can be seen that SE-RCNet’s advantage also lies in its ability to recognize different types of fall behaviors. By analyzing ten different fall behaviors in the K-band ultra-wideband radar fall detection spectrum dataset, SE-RCNet can more accurately distinguish these complex action types. This further demonstrates the model’s superior performance in handling diverse data and recognizing complex behavior patterns.

#### 4.2.2. Comparison of Fusion Methods

In this section, we evaluate the fine-tuning effect of using an adaptive weighted fusion network for radar spectrum feature extraction and action recognition tasks, based on the initial classification results of the SE-RCNet network, we compared the performance of the adaptive weighted fusion network with that of the decision fusion network. We selected accuracy, precision, recall, and F1-score as the evaluation metrics. Below are the detailed experimental results and analysis.

From Table 7, it can be seen that the adaptive weighted fusion network performs excellently across multiple evaluation metrics. It achieved an accuracy of 0.975, a precision of 0.973, a recall of 0.973, and an F1-score of 0.972. Compared to the highest recognition results from individual spectrograms, these metrics improved by 2.08%, 1.87%, 2.94%, and 2.40%, respectively. This indicates that the network not only possesses high accuracy but also exhibits outstanding balance across all performance metrics. As shown in Figure 9, after adaptive weighted fine-tuning, only a small amount of confusion occurred between falling at a 45-degree angle to the left and falling at a 45-degree angle to the right. All other action types achieved 100% recognition.

To further verify the effectiveness of the adaptive weighted fusion network, we conducted comparative experiments on public datasets. The results demonstrated in the confusion matrix of the fusion method on the public dataset in Figure 10 indicate that the adaptive weighted fusion network still outperforms traditional decision fusion networks on the public dataset. As shown in Table 8, the adaptive weighted fusion network achieved an accuracy of 0.764, a precision of 0.768, a recall of 0.769, and an F1-score of 0.768 on the public dataset, while the traditional decision fusion network scored 0.689, 0.688, 0.691, and 0.689 on the respective metrics.

These results indicate that the adaptive weighted fusion network not only performs exceptionally well on self-built datasets but also demonstrates superior performance on public datasets. This can be mainly attributed to its ability to dynamically adjust the weights of sub-models. By adaptively adjusting the weights of each sub-model, the network can optimize the contribution of each sub-model in the final decision based on the current context and characteristics of the input data. This approach allows it to better adapt to different environments and data distributions, thereby improving the overall robustness and generalization capability of the model.

Overall, the adaptive weighted fusion network demonstrates high adaptability and stability in handling radar spectrum feature extraction and motion recognition tasks. Its dynamic adjustment mechanism enables it to maintain efficient recognition performance even in complex and variable environments.

## 5. Conclusions

This paper proposes a novel fall behavior recognition network, SE-RCNet, which significantly enhances recognition performance by integrating residual connections and Squeeze-and-Excitation (SE) modules. Comparative experiments show that SE-RCNet achieved average F1-scores of 94.0%, 94.3%, and 95.4% when processing three types of radar maps, significantly outperforming existing deep learning models.

Additionally, to further improve recognition accuracy and effectively utilize radar data, this paper introduces an adaptive weighted fusion method. This method innovatively adjusts the weight proportions dynamically based on the actual accuracy of each radar map, overcoming the limitations of traditional decision fusion strategies in radar map weight allocation. Experimental results demonstrate that this fusion method not only outperforms traditional single radar map recognition methods but also achieves an F1-score of 98.1% on our custom dataset with SE-RCNet, surpassing the decision fusion method. This indicates that the method can effectively address false positives and false negatives in fall recognition.

To verify the generalization capability of the SE-RCNet model and its adaptive weighted fusion method, we conducted evaluations on a public fall recognition dataset. The results show that the model and fusion method perform better on this dataset compared to traditional methods, confirming their high stability and reliability across different environments when compared to other network models and fusion algorithms.

However, in real life, human behavior is continuous and complex, not just simple, isolated actions. Current fall detection systems typically can only recognize single, obvious fall events, which limits their effective use in real-world environments. While our adaptive weighted fusion method can significantly improve the model’s recognition accuracy, it also presents challenges in terms of model and algorithm deployment, potentially reducing real-time recognition. However, with the continuous improvement in hardware performance and the emergence of hardware acceleration technologies (such as GPUs and FPGAs) and the deployment of edge computing. For the challenges posed by obstacles and electromagnetic interference, which can degrade radar system performance, our network and fusion method are specifically designed to improve recognition accuracy under such challenging conditions. By applying the adaptive weighted fusion algorithm to the results of the three radar maps, we enhance the system’s adaptability and robustness across various environments. Specifically, the adaptive weighted fusion algorithm can dynamically adjust the weight parameters of each model according to different environmental conditions, thereby achieving higher recognition performance. Therefore, future work will focus on recognizing continuous behavioral actions to enhance the general applicability of fall prevention systems and reduce misrecognition rates. Additionally, we will strive to improve the system’s ability to recognize human behavior in various interference environments.

## Figures and Tables

**Figure 1 sensors-24-05294-f001:**
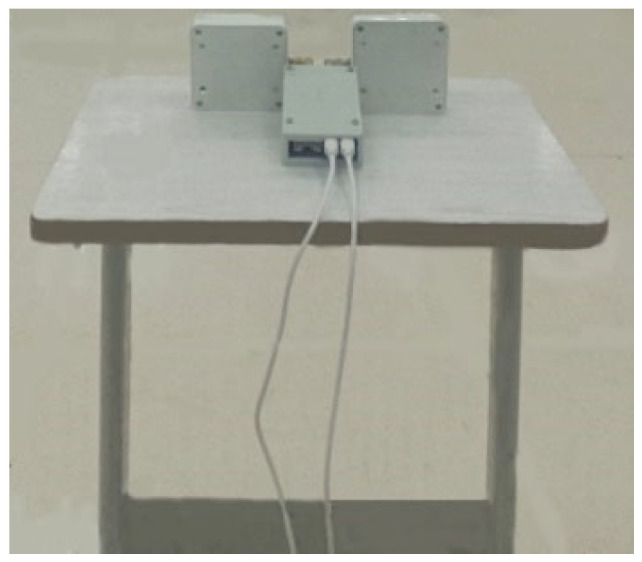
PulsOn 440 radar module.

**Figure 2 sensors-24-05294-f002:**
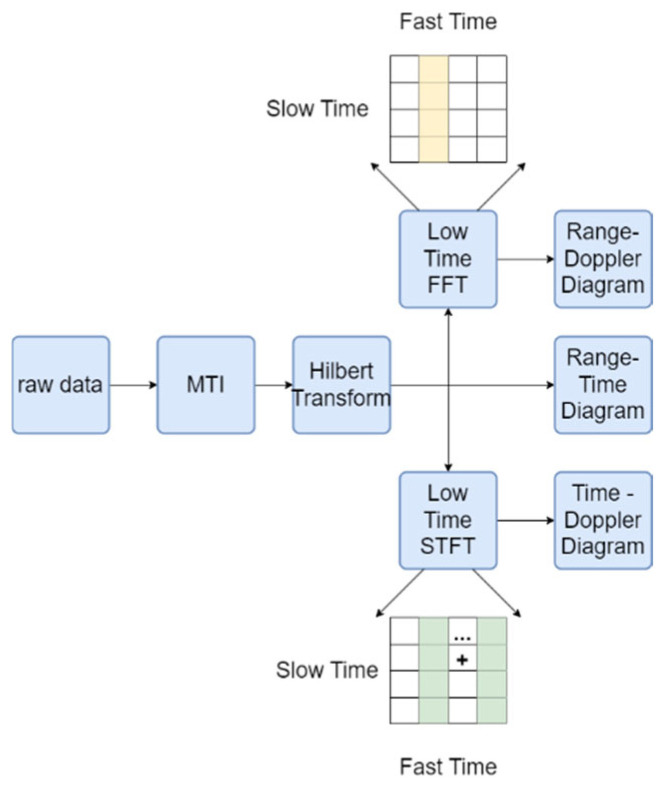
Diagram of Graph Generation.

**Figure 3 sensors-24-05294-f003:**
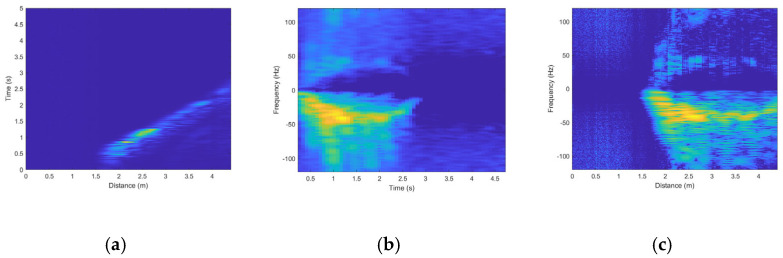
Three Radar Spectrograms for Walking Away from the Radar. (**a**) Range-Time Diagram. (**b**) Time-Doppler Diagram. (**c**) Range-Doppler Diagram.

**Figure 4 sensors-24-05294-f004:**
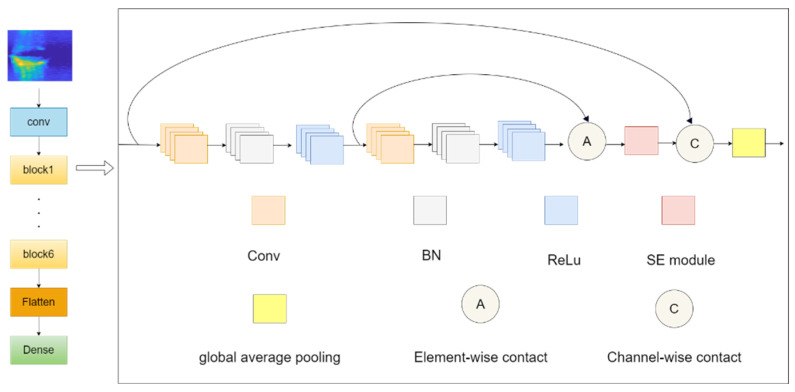
SE-RCNet Model Architecture Diagram.

**Figure 5 sensors-24-05294-f005:**
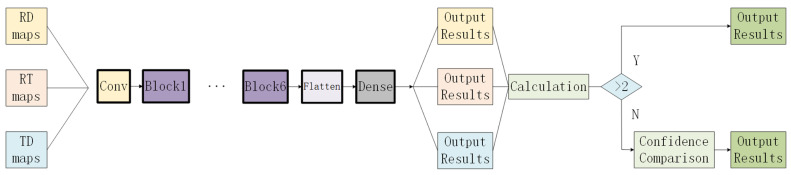
Decision Fusion Diagram.

**Figure 6 sensors-24-05294-f006:**
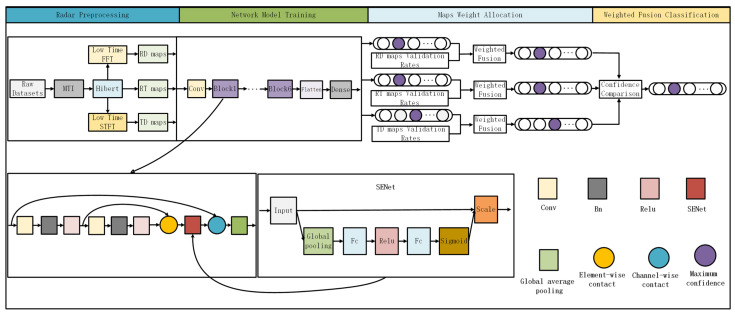
Adaptive Weighted Fusion Architecture Based on SE-RCNet.

**Figure 7 sensors-24-05294-f007:**
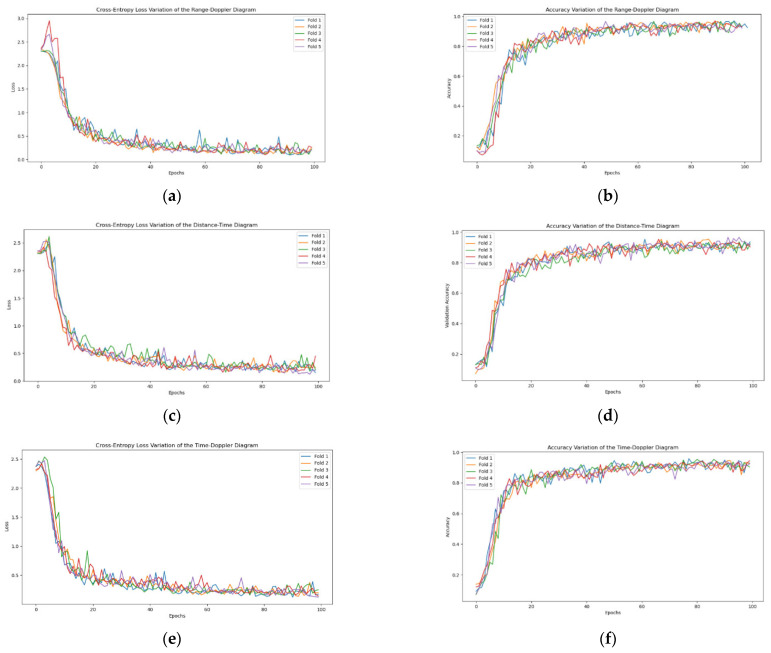
The accuracy and Cross-Entropy Loss Curves of Single-Spectrogram Detection for 5-Fold Cross-Validation over the Number of Training Epochs. (**a**) Cross-entropy loss variation in the range–distance diagram. (**b**) Accuracy variation in the range–distance diagram. (**c**) Cross-entropy loss variation in the distance–time diagram. (**d**) Accuracy variation in the distance–time diagram. (**e**) Cross-entropy loss variation in the time–distance diagram. (**f**) Accuracy variation in the time–distance diagram.

**Figure 8 sensors-24-05294-f008:**
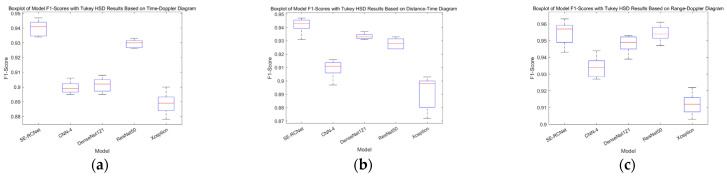
Boxplot of Model F1-Scores on Three Types of Spectrograms. (**a**) Boxplot based on time–doppler diagram. (**b**) Boxplot based on distance–time diagram. (**c**) Boxplot based on range–doppler diagram.

**Figure 9 sensors-24-05294-f009:**
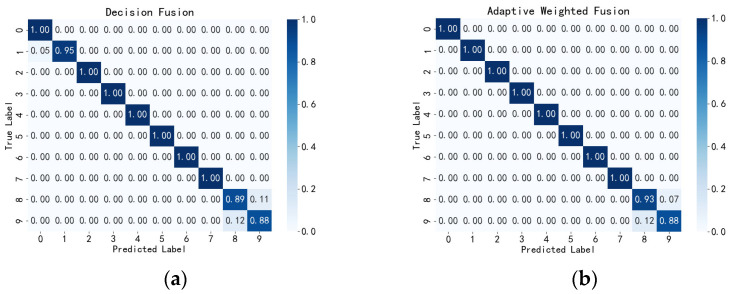
Confusion Matrix of Fusion Methods in the Self-Built Dataset. (**a**) Decision Fusion Confusion Matrix Based on the Self-Built Dataset. (**b**) Adaptive Weighted Fusion Confusion Matrix Based on the Self-Built Dataset.

**Figure 10 sensors-24-05294-f010:**
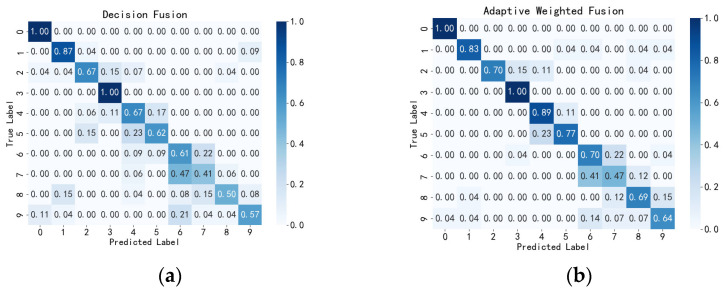
Confusion Matrix of Fusion Methods in the Public Datasets. (**a**) Decision Fusion Confusion Matrix Based on the Public Dataset. (**b**) Adaptive Weighted Fusion Confusion Matrix Based on the Public Dataset.

**Table 1 sensors-24-05294-t001:** Radar Parameter Settings for Self-Built Dataset.

Radar Model	PulsOn 440
pulse repetition frequency center frequency frequency band range sampling frequency	240 Hz 4.3 GHz 3.1~4.8 GHz 16.387 GHz
sampling time	5 s

**Table 2 sensors-24-05294-t002:** Basic Information of Participants.

Subject ID	Gender	Height	Weight	Subject ID	Gender	Height	Weight
1	F	165	54	6	M	175	73
2	M	170	71	7	F	168	59
3	M	168	62	8	M	169	67
4	M	172	65	9	M	170	66
5	M	178	81				

**Table 3 sensors-24-05294-t003:** Description of Actions in the Self-Built Dataset.

Number	Behavior Type	Description
0	Sitting down	The subject sits down from a standing position
1	Falling while sitting/standing	The subject falls while at9tempting to sit or stand
2	Bending to pick something up	The subject bends over to pick up an object from the ground
3	Falling sideways to the radar	The subject falls sideways relative to the radar
4	Standing up	The subject stands up from a seated position
5	Falling backwards to the radar	The subject falls backwards relative to the radar
6	Walking towards the radar	The subject walks towards the radar
7	Walking away from the radar	The subject walks away from the radar
8	Falling towards the radar at a 45-degree angle to the right	The subject falls towards the radar at a 45-degree angle to the right
9	Falling towards the radar at a 45-degree angle to the left	The subject falls towards the radar at a 45-degree angle to the left

**Table 4 sensors-24-05294-t004:** Introduction to Public Human Fall Datasets.

Action Number	Action Type
0	Tripped while going upstairs
1	Tripped while going downstairs
2	Slipped while sitting down backward
3	Fell while standing up from sitting
4	Fainted with back facing the radar
5	Slipped with back facing the radar
6	Fell at 45 degrees to the right facing the radar
7	Fell at 45 degrees to the left facing the radar
8	Fainted facing the radar
9	Fell facing the radar

**Table 5 sensors-24-05294-t005:** Cross-Validation Performance on Custom Spectrograms.

Model	Accuracy			Precision			Recall			F1-Score		
	TD	TR	RD	TD	TR	RD	TD	TR	RD	TD	TR	RD
SE-RCNet	0.933	0.941	0.960	0.935	0.942	0.955	0.945	0.944	0.953	0.940	0.943	0.954
CNN-4	0.898	0.914	0.931	0.898	0.914	0.939	0.894	0.904	0.929	0.896	0.906	0.933
DenseNet121	0.888	0.930	0.949	0.891	0.931	0.949	0.932	0.928	0.947	0.908	0.929	0.948
ResNet50	0.930	0.936	0.952	0.935	0.935	0.956	0.936	0.939	0.950	0.936	0.937	0.953
Xception	0.875	0.884	0.914	0.885	0.894	0.915	0.894	0.888	0.909	0.889	0.891	0.911

**Table 6 sensors-24-05294-t006:** Cross-Validation Performance on Public Spectrograms.

Model	Accuracy			Precision			Recall			F1-Score		
	TD	TR	RD	TD	TR	RD	TD	TR	RD	TD	TR	RD
SE-RCNet	0.575	0.698	0.665	0.570	0.722	0.674	0.562	0.721	0.678	0.566	0.721	0.676
CNN-4	0.550	0.676	0.620	0.549	0.689	0.640	0.530	0.695	0.604	0.555	0.692	0.607
DenseNet121	0.510	0.669	0.559	0.505	0.690	0.632	0.515	0.681	0.604	0.507	0.686	0.607
ResNet50	0.548	0.689	0.663	0.518	0.689	0.671	0.536	0.695	0.663	0.524	0.692	0.667
Xception	0.330	0.512	0.495	0.325	0.551	0.583	0.326	0.557	0.539	0.325	0.554	0.553

**Table 7 sensors-24-05294-t007:** Comparison of Fusion Method Results in the Self-Built Dataset.

	Accuracy	Precision	Recall	F1-Score
TD Diagram (Before Fusion)	0.937	0.933	0.938	0.935
TR Diagram (Before Fusion)	0.945	0.941	0.943	0.942
RD Diagram (Before Fusion)	0.963	0.963	0.953	0.958
Adaptive Weighted Fusion	0.983	0.981	0.981	0.981
Improvement (Adaptive Weighted Fusion)	2.08%	1.87%	2.94%	2.40%
Decision Fusion	0.975	0.973	0.973	0.972
Improvement (Decision Fusion)	1.25%	1.04%	2.10%	1.46%

**Table 8 sensors-24-05294-t008:** Comparison of Fusion Method Results in Public Datasets.

Fusion Methods	Accuracy	Precision	Recall	F1-Score
Adaptive Weighted Fusion	0.764	0.768	0.769	0.768
Decision Fusion	0.689	0.688	0.691	0.689

## Data Availability

The public datasets used in this study are available from https://radars.ac.cn/web/data/getData?newsColumnId=878070cd-079e-454e-a14f-ebccc8cb5ba0 (accessed on 26 May 2024). The self-built dataset presented in this study is available from the corresponding author upon reasonable request.

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
