# Peer review of "Human Fall Detection with Ultra-Wideband Radar and Adaptive Weighted Fusion"

_sensors, 2024, doi:10.3390/s24165294_

Round 1
Reviewer 1 Report
Comments and Suggestions for Authors
The suggested way to use radars for indoor human detection is interesting and has a high potential to be implemented. The used processing technique seems adequate. The experimental results are good. I recommend publishing after a minor revision:
1. Please avoid using abbreviations in the title. I know that "UWB" is well recognizable, but this abbreviation without decoding can be confusing for some readers.
2. In radar parameter list substitute "hz" with "_Hz".
3. Please, indicate radar power to underline that the power level is safe for humans and satisfies FDA requirements.
Author Response
|
Comments 1: Please avoid using abbreviations in the title. I know that "UWB" is well recognizable, but this abbreviation without decoding can be confusing for some readers.
|
|
Response 1:Thank you very much for your suggestion. Indeed, such undecoded abbreviations may confuse some readers who are not specialized in this field. Therefore, I have made the corresponding changes to the manuscript, renaming the title to "Human Fall Detection with Ultra-Wideband Radar and Adaptive Weighted Fusion" to ensure it is understandable to a broader audience. |
|
Comments 2: In radar parameter list substitute "hz" with "_Hz". |
|
Response 2: Thank you very much for your reminder. I have now corrected all the incorrect forms to better align with academic standards.
|
|
Comments 3: Please, indicate radar power to underline that the power level is safe for humans and satisfies FDA requirements. |
|
Response 3:Thank you for your question. I have added more information about the radar equipment and cited the device documentation to demonstrate that the power levels used in our experimental conditions comply with FDA requirements and are safe for humans. The specific content has been supplemented in orange font on the third page of the manuscript. |
Reviewer 2 Report
Comments and Suggestions for Authors
The paper calls: "UWB Radar Human Fall Type Recognition Based on Adaptive Weighted Fusion" and concerned of novel approach recognizing various types of falls by innovative SE-RCNet network. The advantage of article actual and original topic of research and long references list (20 ones).
However reviewer have some questions:
1. How resolution of radar affect on recognition efficiency and accuracy?
2. Which race of object were used? How race of object affect on detection algorithm efficiency?
3. Is it possible of using such algorithm in human security in technological process applications?
Author Response
|
Comments 1: Please avoid using abbreviations in the title. I know that "UWB" is well recognizable, but this abbreviation without decoding can be confusing for some readers.
|
|
Response 1:Thank you very much for your suggestion. Indeed, such undecoded abbreviations may confuse some readers who are not specialized in this field. Therefore, I have made the corresponding changes to the manuscript, renaming the title to "Human Fall Detection with Ultra-Wideband Radar and Adaptive Weighted Fusion" to ensure it is understandable to a broader audience. |
|
Comments 2: In radar parameter list substitute "hz" with "_Hz". |
|
Response 2: Thank you very much for your reminder. I have now corrected all the incorrect forms to better align with academic standards.
|
|
Comments 3: Please, indicate radar power to underline that the power level is safe for humans and satisfies FDA requirements. |
|
Comments 1: How resolution of radar affect on recognition efficiency and accuracy?
|
|
Response 1:Thank you for your question. Regarding your inquiry about how radar resolution affects recognition efficiency and accuracy, we have indeed found in our research that a higher Pulse Repetition Frequency (PRF) can measure the target's velocity more accurately but will reduce the maximum detection range. Additionally, the sampling frequency significantly impacts the time resolution of the radar signal. The higher the sampling frequency, the more details the radar can capture in the signal, which is crucial for improving detection and recognition accuracy. Therefore, we chose a high sampling frequency in our experiment to ensure that the radar can accurately capture fast and subtle movements. In our experiment, we used the PulsOn 440 radar module from Time Domain. This UWB radar has a center frequency of 4.3 GHz and a frequency range of 3.1 to 4.8 GHz. The sampling frequency is 16.387 GHz. We set the PRF(Pulse Repetition Frequency) to 240 Hz, enabling a detection range of approximately 5.3 meters. This setup ensures that the radar can detect human movements within its range while achieving higher velocity resolution, allowing it to capture finer movements. This is crucial for accurately identifying human positions and movements. The specific content has been supplemented in green font on the third page of the manuscript. |
|
Comments 2: Which race of object were used? How race of object affect on detection algorithm efficiency? |
|
Response 2: Thank you for your question. The basic information of the participants is shown in Table 2. Due to the inclusion of participants of different genders (male and female) and varying body profiles (height and weight), although they perform the same actions, the range of their movements is not entirely identical. Generally, the larger the movement amplitude of the experimental subjects, the more distinct the human movement information received by the radar, making recognition easier and the recognition rate higher. Conversely, if the movement amplitude is small, it becomes easier to confuse actions that have subtle differences, thereby reducing the recognition rate. This diversity allows us to simulate a wide range of human movements, ensuring that the collected data covers a broad spectrum of real-world scenarios. This helps to improve the applicability and reliability of the algorithm across different populations, thereby enhancing its accuracy and robustness in behavior classification and recognition. Thank you for your attention and valuable suggestions on our research. The specific content has been supplemented in green font on the third page of the manuscript.
|
|
Comments 3: Is it possible of using such algorithm in human security in technological process applications? |
|
Response 3:Thank you for your question. Regarding the feasibility of applying this algorithm in practical scenarios, I believe it is possible. Our experimental results show that the adaptive weighted fusion algorithm significantly improves recognition performance. However, the current limitations of deploying deep learning models on hardware devices make implementation challenging. With advancements in technology and improvements in hardware performance, the capacity to deploy larger models on devices and achieve real-time recognition will increase. Therefore, I believe that this algorithm will be highly beneficial in enhancing recognition accuracy in future practical applications. The specific content has been supplemented in green font on the 15th page of the manuscript. |